# Potential drivers for schistosomiasis persistence: Population genetic analyses from a cluster-randomized urogenital schistosomiasis elimination trial across the Zanzibar islands

Tom Pennance[1,2], M. Inês Neves[2,3], Bonnie L. Webster[1,2], Charlotte M. Gower[2,3], Stefanie Knopp[1,4,5], Iddi Simba Khamis[6], Shaali M. Ame[7], Said M. Ali[7], Muriel Rabone[1,2], Aidan Emery[1,2], Fiona Allan[1,2], Mtumweni Ali Muhsin[6], Khamis Rashid Suleiman[7], Fatama Kabole[6], Martin Walker[2,3], David Rollinson[1,2], Joanne P. Webster[2,3]*

1 Wolfson Wellcome Biomedical Laboratories, Department of Life Sciences, The Natural History Museum, London, United Kingdom, 2 London Centre for Neglected Tropical Disease Research (LCNTDR), London, United Kingdom, 3 Department of Pathobiology and Population Sciences, Royal Veterinary College, University of London, London, United Kingdom, 4 Swiss Tropical and Public Health Institute, Basel, Switzerland, 5 University of Basel, Basel, Switzerland, 6 Neglected Diseases Programme, Ministry of Health, Zanzibar, United Republic of Tanzania, 7 Public Health Laboratory—Ivo de Carneri, Pemba, United Republic of Tanzania

* jowebster@rvc.ac.uk

**Editor:** jong-Yil Chai, Seoul National University College of Medicine, REPUBLIC OF KOREA

**Data Availability Statement:** All remaining miracidial samples on Whatman FTA cards and

## Abstract

The World Health Organization's revised NTD Roadmap and the newly launched Guidelines target elimination of schistosomiasis as a public health problem in all endemic areas by 2030. Key to meeting this goal is elucidating how selective pressures imposed by interventions shape parasite populations. Our aim was to identify any differential impact of a unique cluster-randomized tri-armed elimination intervention (biannual mass drug administration (MDA) applied alone or in association with either mollusciciding (snail control) or behavioural change interventions) across two Zanzibarian islands (Pemba and Unguja) on the population genetic composition of *Schistosoma haematobium* over space and time. Fifteen microsatellite loci were used to analyse individual miracidia collected from infected individuals across islands and intervention arms at the start (2012 baseline: 1,522 miracidia from 176 children; 303 from 43 adults; age-range 6–75, mean 12.7 years) and at year 5 (2016: 1,486 miracidia from 146 children; 214 from 25 adults; age-range 9–46, mean 12.4 years). Measures of genetic diversity included allelic richness (Ar), Expected (He) and Observed heterozygosity (Ho), inbreeding coefficient ($F_{ST}$), parentage analysis, estimated worm burden, worm fecundity, and genetic sub-structuring. There was little evidence of differential selective pressures on population genetic diversity, inbreeding or estimated worm burdens by treatment arm, with only the MDA+snail control arm within Unguja showing trends towards reduced diversity and altered inbreeding over time. The greatest differences overall, both in terms of parasite fecundity and genetic sub-structuring, were observed between the islands, consistent with Pemba's persistently higher mean infection intensities compared to

associated DNA extractions have been bio-banked within the Schistosomiasis Collection at the Natural History Museum (SCAN), which can be found and accessed at: https://www.nhm.ac.uk/our-science/our-work/sustainability/schistosomiasis-collection.html and https://scan.myspecies.info/ The data that support the epidemiological findings of this study are openly available in ClinEpiDB. The dataset "Study: SCORE Zanzibar S. haematobium Cluster Randomized Trial" can be found at: https://clinepidb.org/ce/app/record/dataset/DS_eddb4757ba. Coding, combined with anonymised microsatellite data associated with the representative schistosome samples, are all available and on GitHub at: https://www.github.com/minesneves/Zanzibar_population_genetics.

**Funding:** This study received financial support from the Bill and Melinda Gates Foundation (BMGF), via University of Georgia Research Foundation, Inc (UGARF) for the Schistosomiasis Consortium for Operational Research Evaluation (SCORE) projects (Population Genetics Grant Refs RR374-053/5054146 & RR374-053/4785426; PIs JPW & DR) in association with the Zanzibar Elimination of Schistosomiasis Transmission (ZEST) project (Ref RR374-053/4893206; PIs SK, DR & FK). AE, MR and FE received funding from Wellcome Trust grant 104958/Z/14/Z, "Schistosomiasis Collections at the Natural History Museum (SCAN)". The funders had no role in the study design, data collection and analysis, decision to publish or preparation of the manuscript.

**Competing interests:** The authors have declared that no competing interests exist.

neighbouring Unguja, and within islands in terms of infection hotspots (across three definitions). These findings highlight the important contribution of population genetic analyses to elucidate extensive genetic diversity and biological drivers, including potential gene-environmental factors, that may override short term selective pressures imposed by differential disease control strategies.

**Trial Registration**: ClinicalTrials.gov ISRCTN48837681.

## Author summary

Schistosomiasis is a parasitic disease caused by infection with blood flukes, which leads to acute and chronic pathology in millions of infected individuals, particularly those within the poorest tropical and subtropical regions. In 2012, the World Health Organization (WHO) set the ambitious goals to achieve Elimination of Schistosomiasis as a Public Health Problem (i.e., EPHP, prevalence of heavy infection intensity less than 1% in all sentinel sites) and complete Interruption of Transmission (i.e., IoT reduction of incidence of infection to zero) in selected African regions by 2025. More recently, the revised WHO Neglected Tropical Diseases (NTD)-Roadmap and Schistosomiasis Control and Elimination Guidelines aim to achieve EPHP in all regions by 2030. Here we analysed population genetic data associated with a recent 5-year cluster-randomized trial across Zanzibar (Pemba and Unguja islands) aimed to assess the impact of contrasting interventions to achieve urogenital schistosomiasis elimination. Whilst, consistent with the main trial study, no significant differential impact of interventions on parasite prevalence or intensity was detected, our data suggested that the greatest impact on genetic diversity was within the mass drug administration plus concurrent mollusciding arm. Moreover, analyses revealed significant differences in both the genetic sub-structuring and notably the fecundity of parasites between Pemba and Unguja islands, and within island in relation to some (but not all) persistent hotspots, potentially indicative of genetic and biological factors driving persistence. These findings highlight the important contribution of population genetic analyses to reveal high levels of genetic diversity, biological drivers and potential gene-environmental interactions, in determining infection dynamics and persistence, all of which may serve to limit the impact of differential disease control activities.

## Introduction

A substantial proportion of the world's disease burden is caused by infectious agents that lead to high mortality, morbidity and reduced productivity among many millions of people. Those living in sub-Saharan Africa (SSA) carry a disproportionate disease burden, much of which is caused by parasitic infections. Guided by the Millennium Development Goals and the subsequent Sustainable Development Goals, great progress has been made in reducing the burden of human infectious diseases since 2000 [1]. However, parasites continue to pose an enormous threat to human and animal health.

Schistosomiasis is a major helminthic neglected tropical disease (NTD), second only to malaria amongst parasitic diseases in terms of socioeconomic impact, and estimated to currently infect over 250 million people [2]. Urogenital schistosomiasis, caused by *Schistosoma haematobium*, accounts for over 110 million of these cases of human schistosomiasis in SSA alone [2]. Since 2002, large-scale mass drug administration (MDA) of preventative

chemotherapy with praziquantel (PZQ) has been implemented across much of SSA [3,4]. Morbidity control has been generally successful across many countries [5] and this has helped lead to a revision of the World Health Organization's (WHO's) strategic plan for a vision of "a world free of schistosomiasis" [6,7]. Most recently, the new WHO NTD-Roadmap and Guidelines both outline aims to achieve schistosomiasis elimination as a public health problem (EPHP, i.e. elimination of morbidity where prevalence of heavy infection intensity less than 1% in all sentinel sites) in all endemic countries by 2030, as well as a complete interruption of transmission (IoT, i.e. reduction of incidence of infection to zero) in selected regions by the same point [8–10].

Zanzibar is an Indian Ocean archipelago, the two largest islands of which are Unguja and Pemba, located ~35km off mainland Tanzania, and for which urogenital schistosomiasis has been recognised as a public health problem for almost a century [11]. The relative isolation of these island populations make them particularly suitable for assessing disease control strategies [12–14], complemented by an apparent lack of animal host reservoirs/zoonotic schistosomiasis transmission (but see [15]), and a *S. haematobium* population genetic lineage distinct from that of much of mainland Africa [16,17]. The 'Zanzibar Elimination of Schistosomiasis Transmission' (ZEST) alliance was formed to evaluate contrasting intervention strategies aimed to achieve EPHP from Pemba and complete IoT on Unguja island within five years [18–20]. The cluster randomized trial involved three study arms, implemented across 90 shehias (small administrative regions) within both islands, to assess the differential impact of: (i) biannual MDA alone; (ii) biannual MDA plus snail control (via molluscicide), and (iii) biannual MDA plus human behavioural change [18,19]. Whilst the ZEST project achieved EPHP in most sentinel sites of both islands, transmission was not interrupted, with no statistically significant differences in the impact of the three treatment arms by the end of the trial on *S. haematobium* prevalence or intensity. Furthermore, ongoing transmission in persistent hotspots was maintained across all arms [20–22].

Our aim here was to identify the impact of the tri-armed intervention selective pressures on the *S. haematobium* population genetic compositions over space and time. We predicted that this approach would reveal greater detail of differing control pressures on the biology and fitness of these parasites, and the potential strategies and counterstrategies of the parasites, over and above that of the standard prevalence and intensity measures inherent within classical monitoring and evaluation alone. We also aimed to elucidate whether those 'hotspot' shehias that showed persistent transmission despite control efforts (as defined in detail below), differed in their genetic diversity or composition relative to parasites from responder sites, which may facilitate precision-targeting of future control interventions. Finally, we aimed to identify any inherent differences in *S. haematobium* obtained from across the two islands, potentially indicative of their longer-term lineage histories and selective pressures imposed and how these may relate to current biological drivers of infection persistence.

## Materials and methods

### Ethics statement

Ethical approval for the cluster randomized trial was obtained from the Zanzibar Medical Research Ethics Committee (ZAMREC 0003/Sept/011), the "Ethikkomission beiber Basel" (EKBB) in Switzerland (reference no. 236/11), and the Institutional Review Board of the University of Georgia (project no. 2012-10138-0). The cluster randomized trial is registered at the International Standard Randomised Controlled Trial, Register Number (ISRCTN48837681). Further approval for the population genetic analyses was obtained from the St Mary's Hospital

Local Ethics Research Committee, R&D office, part of the Imperial College, London Research Ethics Committee (ICREC; (EC NO: 03.36. R&D No: 03/SB/033E)).

All aspects of sample collections for the cluster randomized trial were carried out in collaboration with staff of the Neglected Diseases Programme of the Zanzibar Ministry of Health (MoH) in Unguja and the Public Health Laboratory—Ivo de Carneri in Pemba. Shehia and school authorities were informed about the purpose and procedures of the study. Written informed consent was obtained from parents for all minors (children ≤16 years old) prior to recruitment. All participating adults provided their own signed consent. Participation was voluntary and individuals could withdraw or be withdrawn from the study at any time without obligation. All infected participants were offered treatment with praziquantel (40 mg/kg), donated by Merck KgA and facilitated by the WHO and Zanzibar Ministry of Health, as part of the MDA implemented by the Schistosomiasis Control Initiative (SCI) and MoH.

## Shehia selection and intervention trial

The total population size of Zanzibar at 2012 was estimated at around 1.3 million, with 896,721 people residing in Unguja and 406,848 in Pemba. Unguja was composed of six districts further divided into 210 shehias, and Pemba had four districts divided into 121 shehias. Each shehia contains several villages, which varied considerably in population size (ranging from 482–26,275) and household number [23]. As described in detail elsewhere [18,19], 90 shehias (45 per island) were selected as intervention-clusters of the randomized trial, where 15 shehias per island were assigned to receive one of the three treatment arm interventions. Participants were randomly-selected, asked for consent to participate, and enrolled into the study from each school/shehia. Both populations, comparable across Unguja and Pemba, had received extensive MDA before the ZEST trial began in 2012 and thus "baseline" here does not represent PZQ-naïve populations [11] (and as detailed below).

## Sample collection and microsatellite analyses

*S. haematobium* miracidia were collected from infected participants at baseline in January-March 2012 (before the onset of the ZEST interventions), with community-wide praziquantel treatment then implemented in April 2012. Miracidial collections were again collected from participants found infected four years into the ZEST programs in Jan-April 2016 (Year 5), following biannual treatment rounds in each of 2012–2015 (8 prior rounds in total), with CWT treatment provided in May 2016. Mid-morning urine samples were collected from each individual included in the randomized controlled trial and processed the same day, within the framework on the main ZEST programme. Samples were visually inspected for blood (macrohematuria) using a colour chart and for microhaematuria using reagent strips (Hemastix; Siemens Healthcare Diagnostics GmbH, Eschborn, Germany). *Schistosoma haematobium* eggs were identified and quantified by filtering 10ml of urine through a polycarbonate filter (Sterlitech, Kent, United States of America) and subsequently examined under a microscope by experienced laboratory technicians. Eggs were then concentrated from the remainder of all infected urine samples by filtration using a Pitchford funnel, rinsed and transferred into a clean Petri dish containing mineral water and exposed to light to facilitate hatching of miracidia. Individual miracidia were captured in mineral water and pipetted onto Whatman-FTA Indicating Classic cards (Whatman, Part of GE Healthcare, Florham Park, USA) for storage of DNA at room temperature (see [24,25]).

For available samples from each infected participant, the DNA from 24 randomly-selected miracidia (or all miracidia available where <24) was individually alkaline-eluted from the Whatman-FTA Indicating Classic cards (Whatman, Part of GE Healthcare, Florham Park,

USA) as described in [24,25], and each sample was given a unique ID (anonymised personal identifier number). Each individual miracidium was genotyped using two previously developed *S. haematobium* multiplexed microsatellite PCR panels (Panel 1 and Panel 2) as described in [22] [25–27] (Table A in S1 Text). Initially the Panel 1 PCR was conducted on all DNA elutions, in a 96 well plate. Four microlitres of each reaction was visualized on a 2% gel red agarose gel and all positive reactions were cherry picked, using MICROLAB STAR Liquid Handling Workstation (Hamilton Robotics), for further fragment analysis. DNA elutions that gave successful positive panel 1 reactions were then also cherry picked, in a 96 well format, again using the MICROLAB STAR Liquid Handling Workstation (Hamilton Robotics) and analysed using the Panel 2 multiplex microsatellite PCR as described in [22]. This two-step approach was utilized to reduce the processing of samples with no or inadequate DNA, due to collection and processing errors inherent with these methodologies. Positive and negative (no DNA) controls were included with each set of 96 reactions. All positive microsatellite panel 1 reactions were diluted 1 in 10 before being denatured and injected at an optimal speed of 12 seconds into the Applied Biosystems 3130xl DNA analyser for analysis. Genotype data was exported and allele peaks were checked and edited using Geneious 6.1.4 before being placed into amplicon size "bins" and alleles exported for analysis. (Full details of the protocols followed are provided in S1 Text).

## Data analyses

Microsatellite multilocus genotypes (MLGS) for each miracidium were compiled by combining data from microsatellite panels 1 and 2 using R (v.3.2.1.) [28]. Data for each miracidium were associated with data from the parasitological surveys including ID, year of collection (baseline or year 5), whether the individual was a child (6–8 or 9–12 years old) or adult, infection intensity as *S. haematobium* eggs per 10ml of urine, sex, age, and shehia of residence (adult) or schooling (child), island (Pemba or Unguja) and intervention arm (MDA only, MDA + snail control or MDA + behaviour change). Data on MDA coverage, shehia prevalence and infection intensity were available from the main trial [20]. Linear mixed-effects models with random intercepts (using the intensity in eggs per 10ml as the response variable, intervention arm and age as fixed effects, and shehias as random effects) were fitted to the 2012 baseline and 2016 Year 5 follow up data to explore differences in infection intensity between intervention arms, age and sex within and between islands.

## Genetic diversity

Summary statistics for allelic richness (Ar), expected heterozygosity (He) and observed heterozygosity (Ho), which are measures of genetic diversity, and the inbreeding coefficient ($F_{ST}$) for each infra-population (the parasite population of each child or adult at each time point) were created using Powermarker [29] and FSTAT [30]. Allelic richness is suitable for the comparison of samples of different sizes since it rarefies to the smallest sample size in the dataset, and is thus particularly useful for schistosome population genetics where the number of miracidia collected is partly dependent on infection intensity. The alternative approach of excluding small sample size datasets would bias against lighter intensity infections, which are likely following successful PZQ treatment. The inbreeding coefficient is a measure of the degree of similarity between parasites within an individual where a lower value indicates less relatedness. Differences in Ar, He, Ho and $F_{ST}$ between treatment arms and between baseline and Year 5 were investigated using linear regression models implemented in Stata (StataCorp LP, USA), controlling for age and sex of individuals, shehia/school, island and miracidia sample size. Models used robust standard errors to account for clustering of infra-populations/individuals

within shehias. Two approaches to modelling were taken. Since there was minimal evidence of difference in genetic diversity indices between intervention arms at baseline, effects of the different intervention could be seen by direct comparison of the arms in the Year 5 dataset. Following this, the datasets from the two years were combined and change was modelled over time, using the baseline population as the reference population, allowing for interactions between time, island and arm to be investigated. *Post-hoc* testing of marginal means investigated whether there was a change for each of the arm-island populations between their own 2012 baseline and 2016 follow-up collections.

## Parentage analysis, estimates of adult worm burden and fecundity

The estimation of full-relationships between the miracidia of each infra-population was carried out using COLONY version 2.0.6.1 [31]. COLONY implements a maximum likelihood algorithm when comparing different sibship configurations and also allows for genotyping error [32]. Sibship relationships were used to estimate the number of unique female parental genotypes present and the number of miracidia per sibship cluster within each individual host [33,34]. Unique parental genotypes were subsequently related to the minimum probable number of female schistosomes within a host using our recently developed statistical approach [35]. Inferred female worm burdens (where all outputs statistically accounted for the limited sample size of miracidia available), together with their individual infection intensity, were used to estimate the parasite reproductive success, measured as the mean number of eggs per 10 ml urine per female worm [35]. Linear mixed-effects models with random intercepts (using the inferred number of female worms within hosts as the response variable, island and age as fixed effects, and shehias as random effects) were fitted to the 2012 baseline and 2016 Year 5 data to explore differences in inferred numbers of female worms between islands and treatment arms. Linear mixed-effects models with random intercepts (using the intensity of infection as the response variable, inferred number of female worms within hosts, island and age as fixed effects, and shehias as random effects) were also fitted to explore differences in worm fecundity between different islands and between different shehias within the same island.

## Persistent hotspots

There is no general consensus on how to define biological 'hotspots', although definitions which account for the failure to change over time, and/or high intensity of infection, are favoured [22,36,37]. We therefore defined biological hotspots here as either: (a) shehias where prevalence had not declined by at least 20% of its starting prevalence between 2012 baseline and 2016 Year 5 amongst the matched 9–12 year old age group (excluding three shehias with a very low starting prevalence (<3%) where stochastic events could have a large impact on definition due to the small number of infected individuals involved), or (b) shehias where the infection prevalence was $\geq$ 15% in 9–12 year-old schoolchildren in at least one of the three cross-sectional parasitological surveys conducted in primary schools in 2012, 2013 and 2014 [36]. Using definition (a), nine shehias (out of 90) were defined as hotspots; using (b), 22 shehias (out of 90) were defined as hotspots. Our third categorization included shehias categorized as a 'hotspot' under both definitions, (a) and (b). Since all nine shehias defined as hotspots in definition (a) were included within definition (b) as well, our third categorization (c) included a total of nine shehias. Regression models were again used to determine if there were differences in mean allelic richness and inbreeding coefficients within shehias classified as hotspots or not, with robust standard errors to account for the clustering of infra-populations within shehias.

### Population structure by island and persistent hotspot status

To gain further insights into the population structure of the miracidia, the microsatellite data were subjected to Principal Component Analysis (PCA) using the "adegenet" v2.1.1 package [38] in R (version 3.5.2). Each miracidium MLG was assigned to its island of origin and as either from a hotspot versus non-hotspot using the definitions as above.

## Results

Whilst the infected participants matched exactly to those sampled within the main ZEST programme [20,21], and hatching was attempted from all participants identified egg-positive by urine filtration, the total values regarding the number of participant samples obtained for genotyping were lower due to a combination of non-systematic drop-out from collection, lack of urines provided, hatching success, exclusion of baseline samples where fewer than five miracidia per participant were obtained, and/or poor-quality DNA for amplification. From the baseline collections in 2012, MLGs were obtained from 1,825 miracidia (see Tables A-D in S1 Text for details by island and intervention arm, and https://github.com/minesneves/Zanzibar_population_genetics for full dataset and coding), 1,522 of which were from 176 children (mean of 8.7 miracidia per child) and 303 from the 43 adults (mean of 7.06 miracidia per adult). The age of the participants at baseline varied from 6 to 75 years with a mean of 12.7 years (see Table B in S1 Text for details by sex).

The 2016 Year 5 follow-up dataset comprised of MLGs from a total of 1,700 miracidia, 1,486 of which were from 146 children (mean of 10.18 miracidia per child) and 214 from 25 adults (mean of 8.56 miracidia per adult). The age of hosts varied from 9 to 46 years with a mean of 12.4 years, and thus the age range was lower than the baseline collections (and hence whilst statistical analyses within year encompassed the full age spectrum, comparisons between years were restricted to matched 9–12-year-old age groups where applicable).

### Intensity of infection

Within those individuals where both epidemiological and parasitological material for population genetic analyses were obtained, infection intensity was significantly higher on Pemba compared to Unjuga across all age-groups and time points, most notably amongst the children (t = -8.28, $p<0.0001$, Fig 1). For urogenital schistosomiasis, infection intensities can be further classified into 'high', where $\geq$50 eggs/10ml of urine, and 'low' where 1–49 e/10 ml [39]. Thus the prevalence of heavy intensity infections was significantly higher on Pemba compared to Unjuga across all age-groups and time points (z = 5.63, $p<0.0001$). A significantly higher prevalence of heavy intensity infections was also found in adults compared to children of 6–8 years old within Pemba (z = 3.484, p<0.05). Although there was a general decrease in mean intensity of infection with increasing age, significant mean differences were found solely between children in Pemba of 6–8 years old and adults at baseline (t = -4.97, t = -4.97 $p<0.0001$; Fig 1), with even a non-significant upward trend amongst adults within Pemba. There was a significant decrease in infection intensity across all age groups in Pemba by the end of the study compared to baseline (t = -2.40, $p = 0.01$), consistent with the main ZEST findings, but this failed to reach significance in Unguja.

### Genetic diversity

There was no difference in allelic richness between arms at year 1 baseline (MDA + behaviour change vs MDA only $t = 0.67$, $p = 0.51$; MDA + snail control vs MDA only; $t = 1.58$, $p = 0.12$) or in Year 5 (MDA + behaviour change vs MDA only $t = 0.85$, $p = 0.40$; MDA + snail control

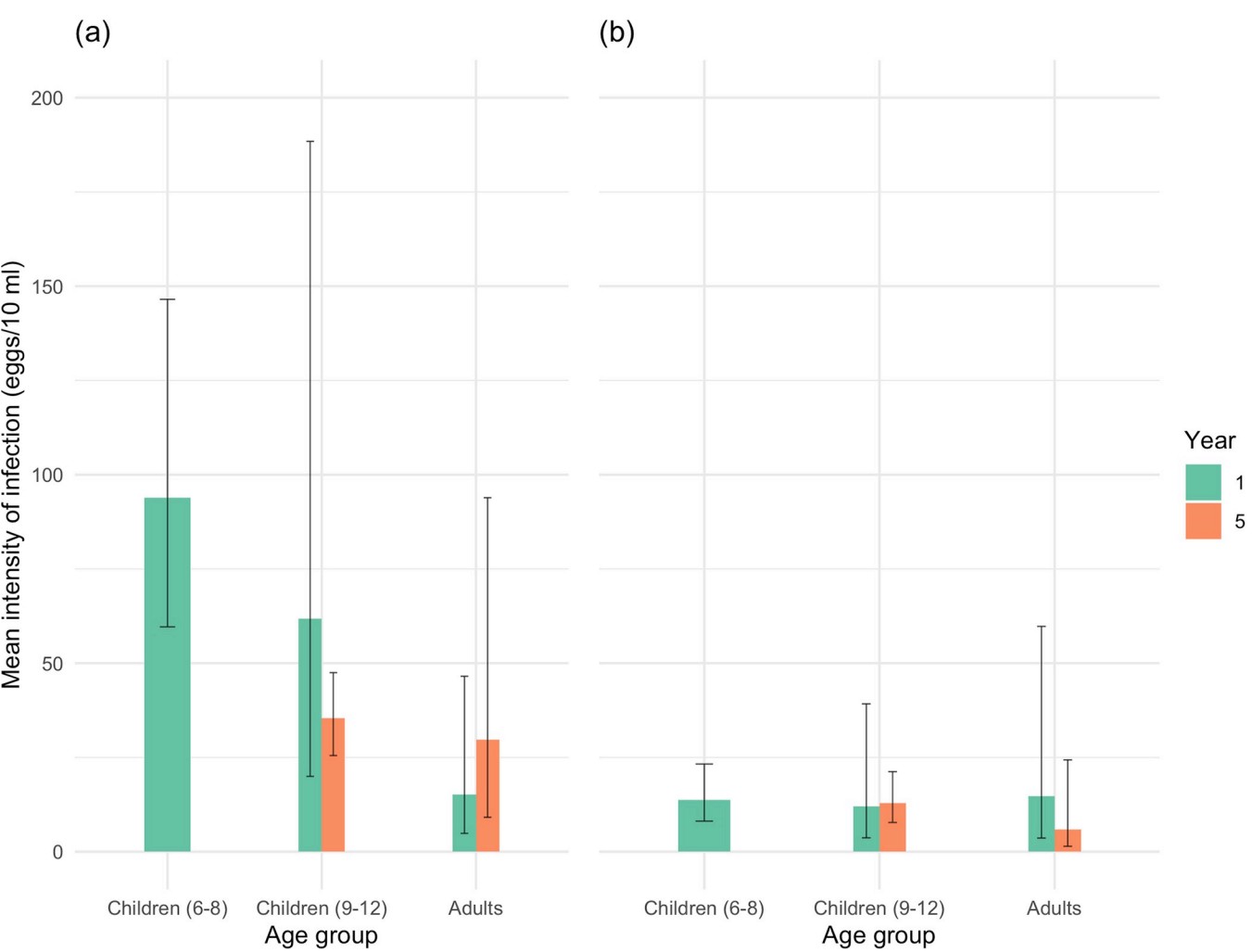

**Fig 1. A-B.** Mean (± 95% confidence interval) intensity of infection (eggs/10ml urine) amongst individuals from different age-groups at baseline and Year 5, pooled across treatment arms but within islands for Pemba (a) and Unguja (b). In Pemba, the baseline data includes 53 children aged 6–8, 31 children aged 9–12 and 26 adults aged 13–75, and the Year 5 data includes 102 children aged 9–12 and 12 adults aged 20–30. Unguja's baseline data consists of 32 children aged 7–8, 57 children aged 9–12 and 18 adults aged 20–54. Unguja's Year 5 data includes 44 children aged 9–12 and 13 adults aged 20–46.

vs MDA only; $t = 0.75$, $p = 0.46$) (see Fig A in S1 Text). Over time, there was no evidence of an overall difference in allelic richness between the baseline and Year 5 collections ($t = 1.44$, $p = 0.16$), nor between the intervention arms (MDA + behaviour vs MDA only ($t = 0.48$, $p = 0.63$); MDA + snail control vs MDA only ($t = 1.62$, $p = 0.11$) or in their allelic diversity at Year 5 in comparison to the baseline MDA only population (Year 5 MDA + behaviour, $t = 0.17$, $p = 0.87$; Year 5 MDA + snail control, $t = 1.20$, $p = 0.23$) (Fig A in S1 Text). However, there was weak evidence of an interaction term between arms and their response to time ($t = -1.77$, $p = 0.08$) and *post-hoc* comparisons of the marginal means demonstrated that this was due to a three-way interaction between time, island and intervention arm ($F_{2, 61} = 3.51$, $p = 0.04$) with a trend towards a reduction in allelic diversity only being present in the Unguja MDA+snail control arm between baseline and Year 5 ($t = -1.88$, $p = 0.07$). Similar results were seen for He and Ho (Figs B and C in S1 Text).

Inbreeding was lower (i.e. outcrossing was higher) in Unguja ($t = -2.03$, $p = 0.05$), although there was no overall change in inbreeding coefficients over time ($t = 1.19$, $p = 0.24$), nor

between the arms with additional interventions in comparison to the MDA only arm (MDA + behaviour, $t = 0.66$, $p = 0.51$; MDA + snail control, $t = 0.82$, $p = 0.41$). However, *post-hoc* comparisons of marginal means demonstrated a difference in the response of the intervention arms to time ($F_{2,72} = 5.19$, $p = 0.008$), when each arm was compared to its own baseline population, with increased outcrossing evident in the snail arms. This was confirmed by analyses of each year independently, where there were no differences between arms or islands in Year 1, but a significantly reduced inbreeding coefficient amongst parasites from the Year 5 MDA +snail arm on Unguja ($t = -0.23$, $p = 0.03$) (Fig D in S1 Text).

### Parentage analysis and estimates of female worm burden

Numbers of female worms per infrapopulation were estimated using sibship reconstruction and our recently developed statistical approach [35]. Despite the higher baseline infection intensities in Pemba island relative to Unjuga, across all arms, there were no significant difference in the mean inferred number of adult worms per infrapopulation (between the islands across both baseline and Year 5 (t = -0.894, p = 0.37) (Fig 2). In both islands, there was a

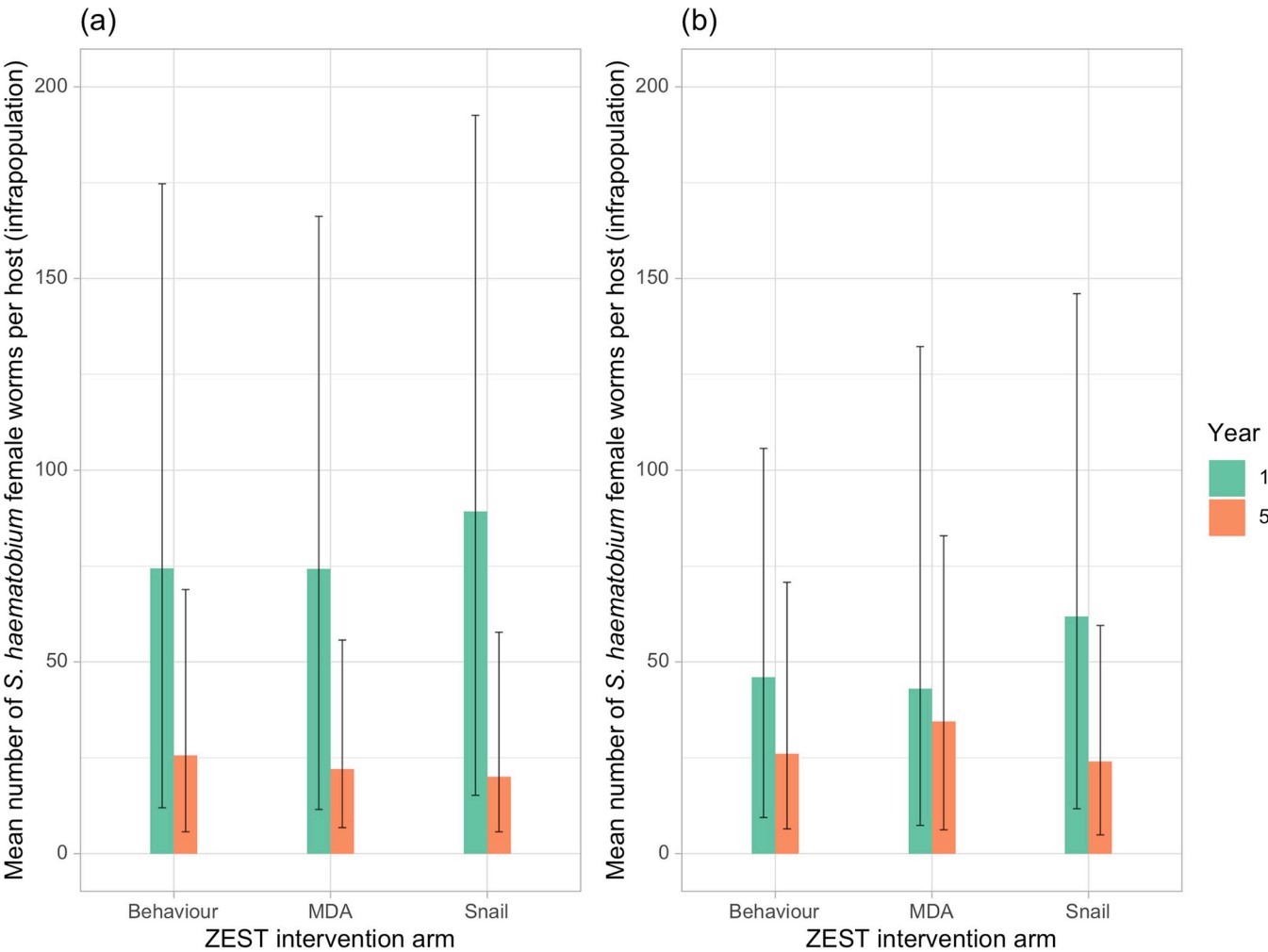

**Fig 2. A-B.** Mean (±95% confidence intervals) estimated number of *Schistosoma haematobium* female worms per host for (a) Pemba and (b) Unguja by treatment arm; where Pemba MDA arm n = 41 baseline, n = 22 year 5; Snail arm n = 40 baseline, n = 55 year 5; Behaviour arm n = 29 baseline, n = 37 Year 5; and where Unjuga MDA arm n = 20 baseline, n = 18 year 5; Snail arm n = 46 baseline; n = 8 year 5; Behaviour arm n = 43 baseline, n = 31 Year 5.

significant decrease in the inferred numbers of female worms from baseline to Year 5 across all arms, indicative of successful control efforts reducing the mean adult worm populations (t = -10.21, p<0.001). However, there were no significant differences in the inferred minimum number of female worms between different treatment arms in both islands at either baseline or Year 5 (t = 0.50, p = 0.62).

## Parasite fecundity

From the estimation of adult worm burden per host, the inferred minimum mean number of eggs produced per female worm (per 10 ml urine) was significantly higher in Pemba than Unguja at baseline (32.9 (13.9–79.6) v 10.5 (2.7–42.2) in Unguja: t = -5.355, p<0.001) and Year 5 (14.6 (7.9–27.1) v 4.4 (1.5–12.8): t = *-5.113, p<0.001*) Fig 3). This is consistent with the observation that infection intensity was consistently higher in Pemba (Fig 1), despite the inferred number of female worms being similar between the two islands (Fig 2). There were not, however, any significant differences in fecundity within islands, in terms of either treatment arms

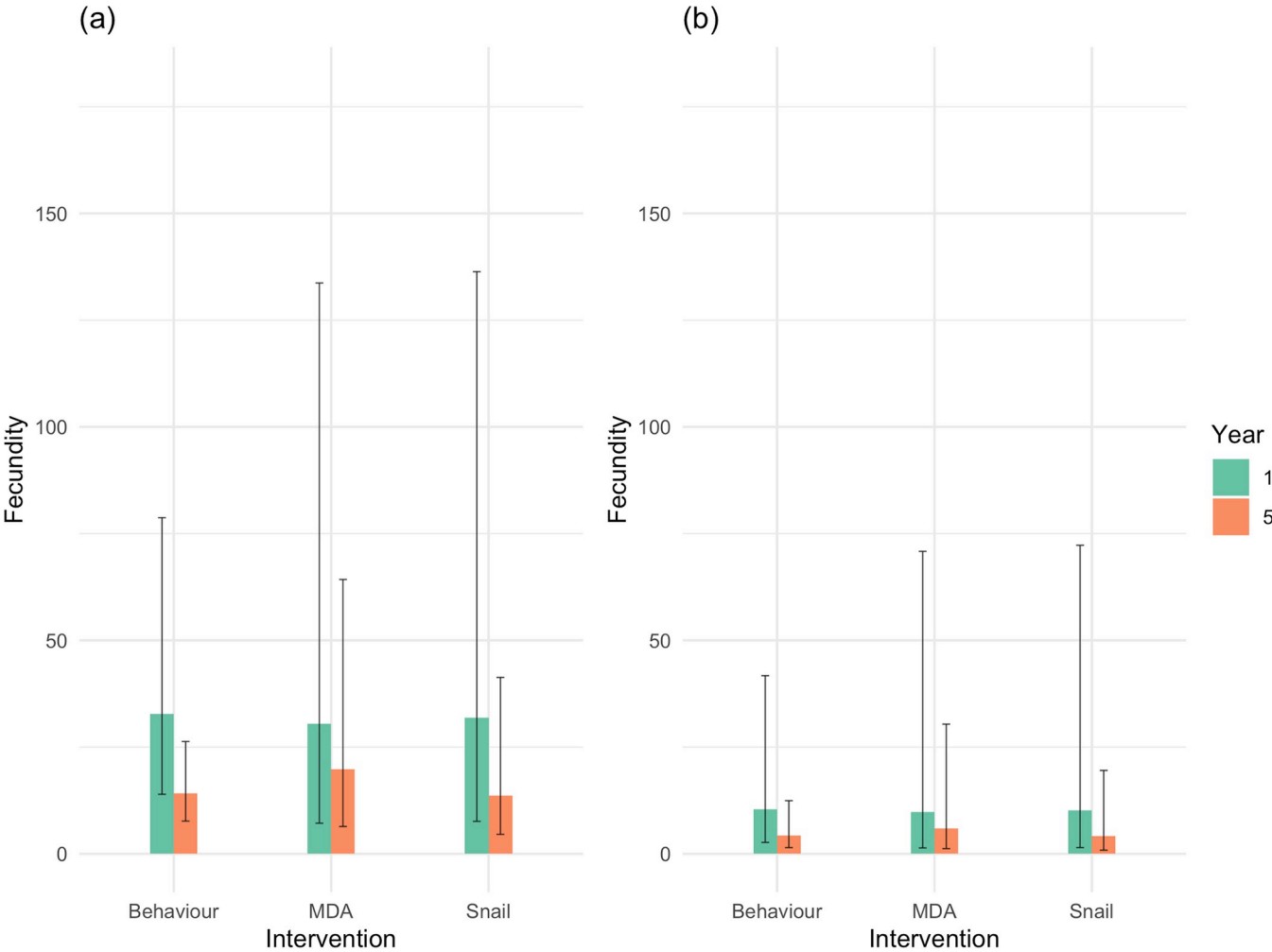

**Fig 3. A-B.** Inferred mean (±95% confidence interval) fecundity (number of eggs per 10ml urine per *Schistosoma haematobium* female worm) in (a) Pemba and (b) Unguja by treatment arm and year; where Pemba MDA arm n = 41 baseline, n = 22 year 5; Snail arm n = 40 baseline, n = 55 year 5; Behaviour arm n = 29 baseline, n = 37 Year 5; and where Unjuga MDA arm n = 20 baseline, n = 18 year 5; Snail arm n = 46 baseline; n = 8 year 5; Behaviour arm n = 43 baseline, n = 31 Year 5.

or years, where all reduced in reponse to control irrespective of treatment arm (Fig 3), although there were significant differences between adults and the 5–8-year-old children at baseline (t = 4.769, p<0.001).

## Persistent hotspots

Fig 4A and 4B shows the mean fecundity (number of eggs per 10 ml urine per *S. haematobium* female worm) for each shehia at baseline and Year 5 estimated from the mixed effects model. Intensity levels at baseline were significantly higher in Pemba hotspots relative to Unjuga hotspots (t = -6.20, $p$<0.001), although there was no evidence of an interaction between islands and hotspots (t = -1.26, $p$ = 0.22). We did not find significant differences in mean intensity levels (t = 0.66, $p$ = 0.514) or fecundity (t = 0.227, $p$ = 0.821) at baseline between individuals in shehias that went on to be designated as persistent hotspots. However, there was a non-significant trend in the inferred numbers of females at baseline between individuals in shehias that wernt on to be considered persistent hotspots (according to definition (b), which also includes all the hotspots in definition (a), t = 1.948, p = 0.057). Across both islands and both years, there was no difference in the mean number of adult worms per infrapopulation estimated from parentage analysis (t = 0.08, $p$ = 0.94) and fecundity (t = 0.77, p = 0.44) in shehias designated as hotspots compared to responders. However, the estimated mean number of eggs per female worm was higher in some hotspots compared to the other shehias in both years (Fig 4A and 4B).

In Pemba, at baseline and Year 5, shehias Kangagani, Uwandani, Vitongoji and Pujini (located in the eastern part of the island), Wambaa (in the western part of the island) and Shumba Viamboni (at north), were found to have mean minimum worm fecundities ranging from 29.8 to 48.5 eggs/10ml urine/female worm, and were considered hotspots by both definitions (Fig 5). In Unguja, parasite mean fecundities ranging from 13.6 to 14.7 eggs/10ml urine/female worm were found in Kitope and Koani, which were also defined as hotspots by both definitions (c). These shehias are located in the centre and northwest of the island (Fig 5).

Parasites also showed a clear genetic distinction between islands (Fig 6A), with also a potential separation by hotspot status within Pemba (Fig 6D and 6E), although with more overlap within Unguja (Fig 6B and 6C).

## Discussion

Our main aim was to elucidate the impact of a tri-armed intervention trial on the population genetics, and how these relate to the transmission dynamics observed, of *S. haematobium* across two islands of the Zanzibar archipelago, thereby helping identify the most effective elimination strategy(ies). We predicted this would reveal greater detail on the impact of differing intervention pressures on the biology and fitness of these parasites, over and above that of the standard epidemiological (prevalence and intensity) measures inherent within classical monitoring and evaluation approaches [40]. We also aimed to elucidate whether those shehias classified as "persistent hotspots" differed in their genetic diversity, composition or apparent parasite fitness which may provide an explanation for their persistence and allow precision targeting of future, and maybe more intense, control interventions. The main ZEST programme reported that IoT was not achieved during the five years on either island, and EPHP was only achieved in Unguja, with infection intensities remaining consistently higher in Pemba. Furthermore, no significant differences were observed between treatment interventions on either island [20,21]. Population genetic analyses here revealed, however, four major additional findings, potentially indicative of gene-environmental-interaction factors at play, all which may present major challenges for reaching the WHO NTD-Roadmap targets:

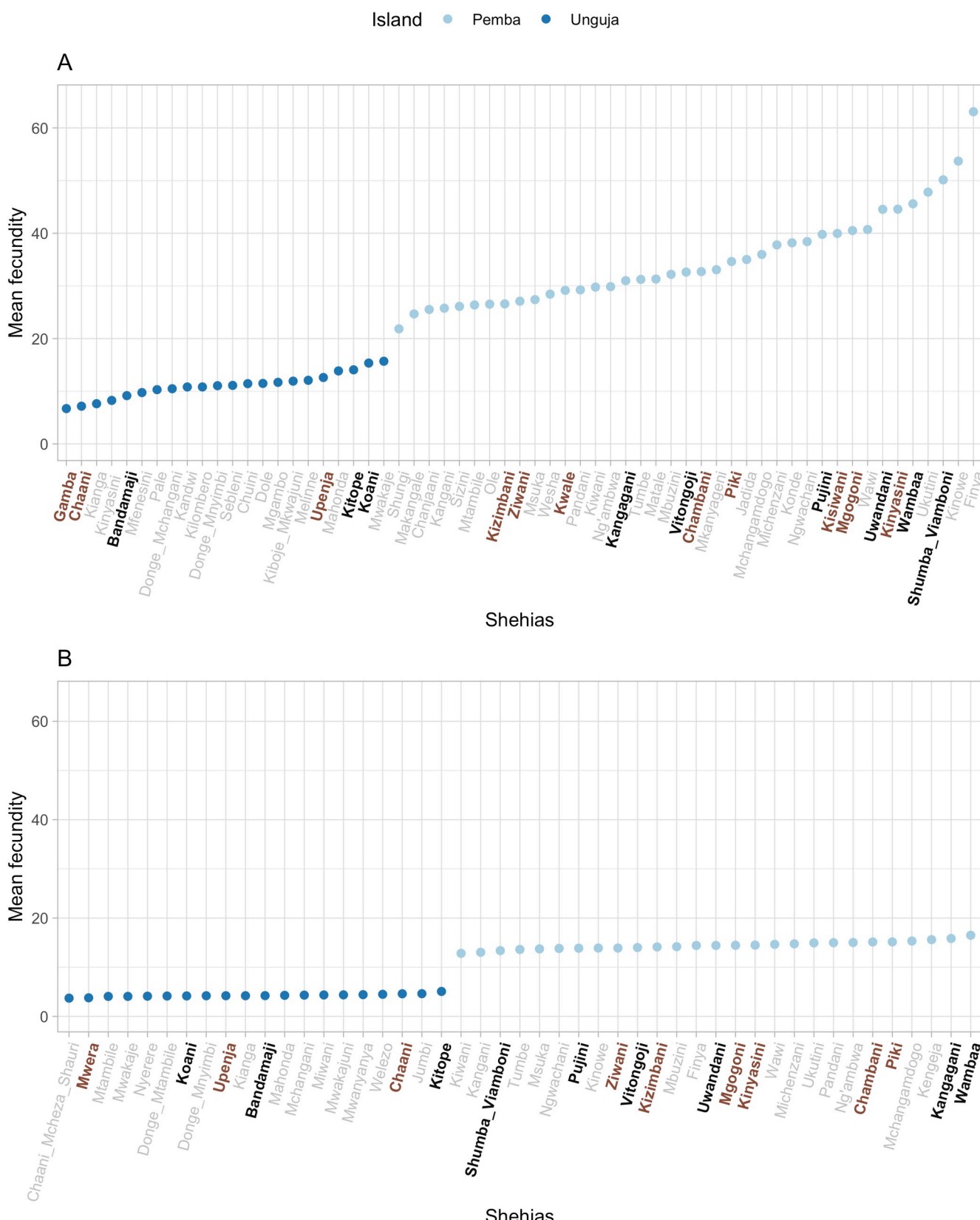

**Fig 4.  A-B.** Mean fecundity of schistosomes (eggs per 10ml urine per female worm) within each shehia at Year 1 baseline (a) and Year 5 follow up (b) estimated using a mixed effects model with shehias as random effects. On the *x*-axis, shehias labelled in brown (bold) correspond to persistent hotspots defined by definition (b) where the infection prevalence was $\geq$ 15% in 9–12 year-old schoolchildren in at least one of the three cross-sectional parasitological surveys conducted in primary schools in 2012, 2013 and 2014 [36]. Shehias labelled in black (bold) correspond to hotspots defined by both definition (b) and definition (a) (i.e., category (c) hotspots, see Material and Methods, *Persistent hotspots*) where prevalence had not declined by at least 20% of its starting prevalence between baseline and Year 5 amongst the matched 9–12-year-old age group. (Error bars not included for visual clarity and because, as a mixed effects model was used, few efficient methodologies to incorporate uncertainty into random effects parameters are available).

(i) There were limited population genetic differences between intervention arms overall within either island, potentially reflective of ongoing high gene flow and refugia [41], as well as the relatively stronger selection pressures imposed already through the longer term disease-control activities across both islands prior to ZEST [11]. Within Unguja, however, a trend towards the greatest reduction in genetic diversity and inbreeding coefficients was observed, potentially indicative of stronger recent selective pressures imposed within the MDA+snail arm over time, relative to the other intervention arms;

(ii) Despite the consistently higher mean infection intensities in Pemba relative to Unguja (Fig 1), the estimated mean numbers of female/adult worm pairs per host were effectively matched across islands and treatment arms (Fig 2). Instead, the mean number of eggs produced, and hence the individual inferred fecundity of each female worm/worm pair, was significantly higher amongst parasites from Pemba than Unguja overall (Fig 3), and most apparent amongst the infected children. Such findings could thereby indicate potential strain differences in terms of fecundity between the islands, and hence also differential biological drivers of persistence;

(iii) Differences in parasite fecundity were also observed by shehia/geographical location within both islands and across intervention arms (Figs 4–6). Whilst again the mean estimated fecundity amongst hotspots (according to definition (b), which also includes all the hotspots in definition (a)) was consistently higher in Pemba than Unguja across both timepoints (Fig 4A and 4B), there was no, however, overall difference in the mean number of adult worms per infrapopulation estimated from parentage analysis in shehias designated as hotspots, compared to responders. Thus it appears that whilst fecundity differences may be a potential contributor to (and characteristic of) the observed hotspots and geographical variability in epidemiological profiles within islands, it is unlikely to be a sole driver.

(iv) Further evidence of potential strain variations amongst parasites by island was revealed by PCA analyses, where despite some overlap, parasites from Pemba and Unguja formed distinct clusters (Fig 6A). In terms of genetic clustering by hotspot, whilst there was overlap throughout Unguja (Fig 6B and 6C), there was some evidence of genetic clustering by hotspot within Pemba (Fig 6D and 6E).

There may be several potential, and likely overlapping, environmental and anthropogenic factors that explain the observed population genetic similarities between the contrasting treatment arms within each island, yet noticeable differences between islands overall, particularly in terms of the consistently higher mean infection intensity and estimated worm fecundity across Pemba relative to Unguja. In terms of the former, as mentioned above, schistosomiasis control activities, and hence longer-term drug-related selective pressures, have been taking place across both islands prior to the commencement of the ZEST clinical trial in 2012 [11]. Indeed, the first morbidity control activities commenced (on Pemba island) in 1986 [42], with both school-based and often community-wide praziquantel treatment ongoing across both islands, albeit intermittently, until 2011 [11]. Thus one could suspect that any subsequent praziquantel pressures, as imposed across all three intervention arms in both islands, may be diluted, with perhaps only the additional pressure of simultaneous mollusciding showing any impact trend. However, treatment pressures prior to ZEST is not sufficient to explain the

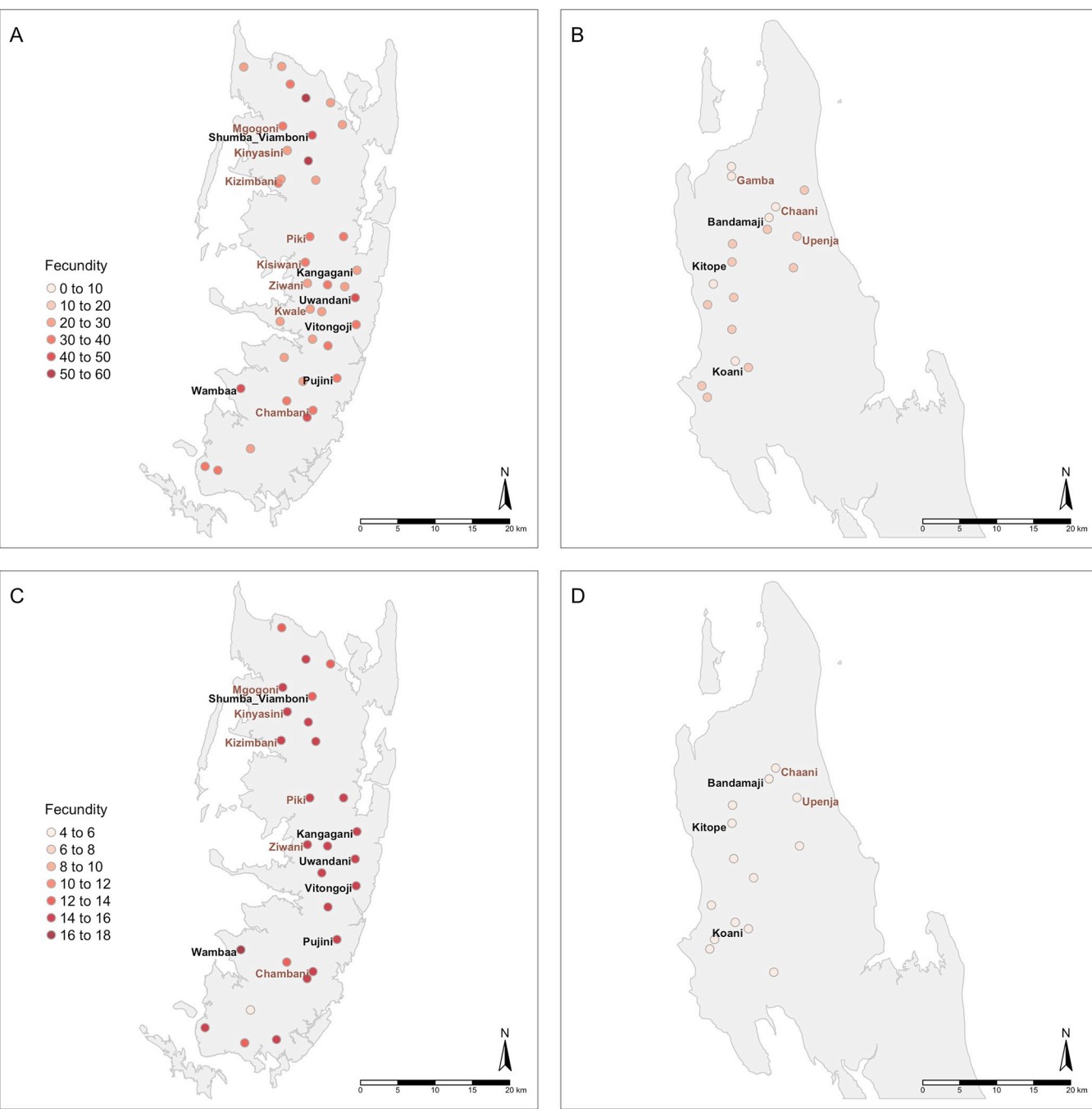

**Fig 5. A–D.** Mean fecundity of schistosomes (eggs per *Schistosoma haematobium* female worm) and location of each shehia in 2012 Year 1 baseline (a and b) and 2016 Year 5 follow up (c and d), in Pemba (a) and (c) and Unguja (b) and (d). Shehias labelled in brown correspond to persistent hotspots by definition (b) and shehias labelled in black correspond to hotspots given the definitions (a) and (b) (see Material and Methods, *Persistent hotspots* for definitions). Note that, whilst the colour schemes are the same in both figures, the scales are different between different years, as fecundity ranged from 6.5–62 in Year 1 and 4–17 in Year 5). The maps were created in RStudio, using the tmap package: https://data.humdata.org/m/dataset/451bdd28-d06d-46ea-91c0-2e081f884395?force_layout=light No changes were made. The License for these data can be found at: https://data.humdata.org/m/dataset/451bdd28-d06d-46ea-91c0-2e081f884395?force_layout=light and are freely-available under the CC BY-IGO license.

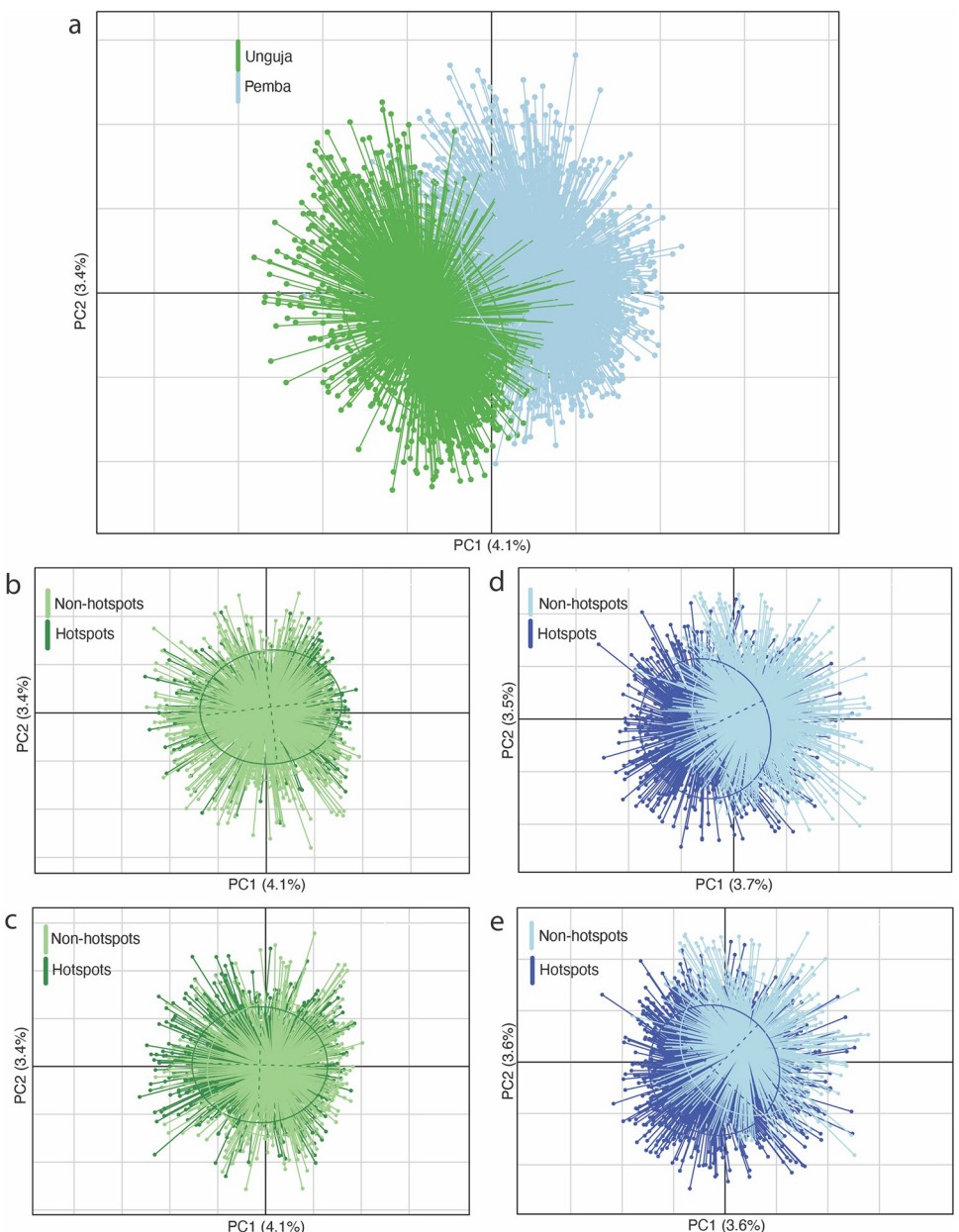

**Fig 6. A-D.** Population structure by island and persistent hotspot status using Principal Component Analysis (PCA) where each point represents an individual miracidium. Each miracidia MLG was assigned as either being from island and/or hotspots versus non-hotspots populations from baseline using the definitions as above. Scale represented as Euclidean distance, d = 0.5. Inertia ellipses indicate the distribution of the individuals from different groups (island or hotspot status). 6b –Unguja hotspot definition A. 6c –Unguja hotspot definition B. 6d –Pemba hotspot definition A. 6e –Pemba hotspot definition B.

significant differences across a range of population genetic indices observed between islands, given that such control interventions were generally matched between islands prior to, and explicitly matched during, ZEST activities [11].

Likewise, localised differences in snail density, and also the presence of a second recently-reported intermediate host species, *B. nasutus*, along the eastern coast of Pemba, [43] could potentially help account for the observed within-island genetic variability and geographic

clustering, and perhaps accentuate between-island variability, irrespective of intervention arm. However, it appears unlikely that snail differences alone could be sufficient to explain the observed between-island structuring, since the major intermediate host species, *Bulinus globosus*, is present across all study regions and across both islands [43].

One could thereby propose instead that one major driver for the apparent genetic and phenotypic (fecundity) strain structuring between the two islands could relate to differential gene flow mediated by both contemporary and historical human movement. Whilst there is free movement and settlement between both islands and mainland Tanzanian, as well as Kenyan communities [14], this is particularly true for Unguja, with its superior ports, in contrast to the more inaccessible Pemba. Unguja is also the largest and most populated island of Zanzibar, with more than 40% of its population living in Stone Town, the administrative and commercial centre of the two islands. Pemba, in contrast, is hillier, wetter and more fertile than Unguja, with farming and fishing as its main income. Furthermore, Pemba is geologically older than Unguja, and whilst both islands have been inhabited since the Paleolithic period, traders began to settle within Pemba at an earlier date than Unjuga, around the 9th or 10th century, albeit with few Europeans entering and indeed between 1964 until the 1980s, Pemba was closed to foreigners [44]. Moreover, though both islands were major centres for the slave trade, Pemba, with more distance to the coast, came to play a greater role in the ongoing illegal slave trade, which has been shown to be associated with differential genomic flow of *Schistosoma mansoni* at least [45]. Such genetic sub-structuring between islands could plausibly be maintained nowadays, despite high levels of movement amongst the, generally less-infected, adult population, as the children will be the key demographic hosts to continue transmission locally. Such parasite genetic and phenotypic differences between islands may well be predicted to impact the effectiveness and sustainability of intervention. The significantly higher mean fecundity levels observed throughout our study in Pemba relative to Unguja indicates a higher parasite fitness in Pemba and suggests that elimination might be harder to achieve.

Geographical variation in helminth fecundity is not without precedent, having been described previously for human roundworm infections and highlighted as an important potential determinant of the success of intervention efforts [46]. In Pemba, poor efficacy of albendazole against *Trichuris trichiura* (whipworm) and hookworm has also been reported [47,48]. Likewise it may also be of interest to note that other endemic diseases, notably lymphatic filariasis, have remained a greater challenge for elimination in Pemba despite successes in Unguja [14].

We fully acknowledge the inherent limitations of the current study, showing data from two key time points rather than annually. Furthermore, when assessing infection intensities and subsequent fitness parameters through molecular analyses, notably that of fecundity, the natural daily variation in egg shedding of *S. haematobium*, as well as the numbers that remain trapped within the host, has to be acknowledged, even when other parameters are controlled for, such as the timing of collection and numbers of miracidia hatched.

## Conclusion

Consistent with the main ZEST programme, our findings indicated little differential impact of the contrasting treatment interventions on the immediate population genetic structure of the circulating parasites (with the possible exception of altered allelic diversity and inbreeding indices over time within the MDA plus mollusciding arm within Unguja alone). Furthermore, our study revealed potential additional challenges to the WHO targets raised by inherent biological differences (notably that of parasite fecundity) and gene-environmental interactions amongst the circulating parasites that may serve to explain the focal nature of schistosome

transmission. Going forward, additional molecular tools, such as whole genome sequencing and SNP analyses, could now be used further to identify differences between the islands, together with any particular strains within 'hotspot' that may have persisted and/or recrudesced over time [41,49]. This should be combined with precision mapping aimed to elucidate the unique ecology, notably in relation to the intermediate hosts present, of these hotspots. Future targeted interventions should thereby be designed in the context of local population genetic and phenotypic data, particularly on factors that relate directly to the fitness and likely resilience of the parasite population to intervention.

## Supporting information

**S1 CONSORT Checklist. CONSORT 2010 checklist of information to include when reporting a randomised trial.**
(DOC)

**S1 Text. Supplementary Information.** Fig A. Adjusted mean ((± 95% confidence interval) allelic richness (Ar) by island, intervention arm and year where where Pemba MDA arm n = 41 baseline, n = 22 year 5; Snail arm n = 40 baseline, n = 55 year 5; Behaviour arm n = 29 baseline, n = 37 Year 5; and where Unjuga MDA arm n = 20 baseline, n = 18 year 5; Snail arm n = 46 baseline; n = 8 year 5; Behaviour arm n = 43 baseline, n = 31 Year 5. Figs B and C. Mean (± 95% confidence interval) (b) Expected Heterozygosity (He) and (c) Observed Heterozygocity (Ho) by island, intervention arm and year where Pemba MDA arm n = 41 baseline, n = 22 year 5; Snail arm n = 40 baseline, n = 55 year 5; Behaviour arm n = 29 baseline, n = 37 Year 5; Fig D. Mean (± 95% confidence interval) inbreeding coefficient (FST) by island, intervention arm and year by island, intervention arm and year where Pemba MDA arm n = 41 baseline, n = 22 year 5; Snail arm n = 40 baseline, n = 55 year 5; Behaviour arm n = 29 baseline, n = 37 Year 5. The inbreeding coefficient is a measure of the degree of similarity between parasites within an individual where a lower value indicates less relatedness.
(DOCX)

## Acknowledgments

We are extremely grateful for the time and effort all participants involved, from the children and adults, their parents, teachers and community leaders. We are extremely grateful to all our miracidial hatching team, including Haji Amer and Juma Kinole Mussa. We also thank Khalfan A. Mohamed for his initial involvement in neglected disease control activities across Zanzibar, and to Julia Llewellyn-Hughes and Lisa Smith in the NHM Molecular laboratories for assistance with microsatellite processing.

## Author Contributions

**Conceptualization:** David Rollinson, Joanne P. Webster.

**Data curation:** Muriel Rabone, Aidan Emery, Fiona Allan.

**Formal analysis:** Tom Pennance, M. Inês Neves, Bonnie L. Webster, Charlotte M. Gower, Martin Walker, Joanne P. Webster.

**Funding acquisition:** David Rollinson, Joanne P. Webster.

**Investigation:** Tom Pennance, Stefanie Knopp, Iddi Simba Khamis, Shaali M. Ame, Said M. Ali, Fiona Allan, Mtumweni Ali Muhsin, Khamis Rashid Suleiman, Fatama Kabole, David Rollinson, Joanne P. Webster.

**Methodology:** Tom Pennance, M. Inês Neves, Bonnie L. Webster, Charlotte M. Gower, Iddi Simba Khamis, Shaali M. Ame, Fiona Allan, Joanne P. Webster.

**Project administration:** Stefanie Knopp, Mtumweni Ali Muhsin, Fatama Kabole, David Rollinson, Joanne P. Webster.

**Resources:** Aidan Emery, Joanne P. Webster.

**Supervision:** Shaali M. Ame, Said M. Ali, Mtumweni Ali Muhsin, Khamis Rashid Suleiman, Fatama Kabole, Martin Walker, David Rollinson, Joanne P. Webster.

**Validation:** Martin Walker, Joanne P. Webster.

**Visualization:** Joanne P. Webster.

**Writing – original draft:** Joanne P. Webster.

**Writing – review & editing:** Tom Pennance, M. Inês Neves, Bonnie L. Webster, Charlotte M. Gower, Stefanie Knopp, Muriel Rabone, Aidan Emery, Fiona Allan, Fatama Kabole, Martin Walker, David Rollinson, Joanne P. Webster.

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
