## [Decision Letter · Decision Letter 0]

6 Jun 2022

Dear Prof. WEBSTER,

Thank you very much for submitting your manuscript "Potential drivers for schistosomiasis persistence: population genetic analyses from a cluster-randomized urogenital schistosomiasis elimination trial across the Zanzibar islands" for consideration at PLOS Neglected Tropical Diseases. As with all papers reviewed by the journal, your manuscript was reviewed by members of the editorial board and by several independent reviewers. The reviewers appreciated the attention to an important topic. Based on the reviews, we are likely to accept this manuscript for publication, providing that you modify the manuscript according to the review recommendations. 

Your manuscript has been reviewed by two independent reviewers. One reviewer recommended accepting it and the other recommended minor revision. We decided to accept your manuscript after minor revision. Please see the comments and address them carefully by a point-to-point basis and resubmit. Thank you for your interest in PLoS NTD.

Sincerely,

jong-Yil Chai

Associate Editor

Maria Elena Bottazzi

Deputy Editor

Your manuscript has been reviewed by two independent reviewers. One reviewer recommended accepting it and the other recommended minor revision. We decided to accept your manuscript after minor revision. Please see the comments and address them carefully by a point-to-point basis and resubmit. Thank you for your interest in PLoS NTD.

Reviewer's Responses to Questions

**Key Review Criteria Required for Acceptance?**

**Methods**

-Are the objectives of the study clearly articulated with a clear testable hypothesis stated?

-Is the study design appropriate to address the stated objectives?

-Is the population clearly described and appropriate for the hypothesis being tested?

-Is the sample size sufficient to ensure adequate power to address the hypothesis being tested?

-Were correct statistical analysis used to support conclusions?

-Are there concerns about ethical or regulatory requirements being met?

Reviewer #1: see general comments

Reviewer #2: -Are the objectives of the study clearly articulated with a clear testable hypothesis stated? YES

-Is the study design appropriate to address the stated objectives? YES

-Is the population clearly described and appropriate for the hypothesis being tested? YES

-Is the sample size sufficient to ensure adequate power to address the hypothesis being tested? YES

-Were correct statistical analysis used to support conclusions? YES

-Are there concerns about ethical or regulatory requirements being met? NO

**Results**

-Does the analysis presented match the analysis plan?

-Are the results clearly and completely presented?

-Are the figures (Tables, Images) of sufficient quality for clarity?

Reviewer #1: see general comments

Reviewer #2: -Does the analysis presented match the analysis plan? YES

-Are the results clearly and completely presented? MOSTLY

-Are the figures (Tables, Images) of sufficient quality for clarity? YES

**Conclusions**

-Are the conclusions supported by the data presented?

-Are the limitations of analysis clearly described?

-Do the authors discuss how these data can be helpful to advance our understanding of the topic under study?

-Is public health relevance addressed?

Reviewer #1: see general comments

Reviewer #2: -Are the conclusions supported by the data presented? SOME

-Are the limitations of analysis clearly described? SOME

-Do the authors discuss how these data can be helpful to advance our understanding of the topic under study? YES

-Is public health relevance addressed? YES

**Editorial and Data Presentation Modifications?**

Reviewer #1: accept

Reviewer #2: (No Response)

**Summary and General Comments**

Reviewer #1: General comment

The experiment reported in the paper is in my opinion well conducted and the paper is very well written 

I think the main aim of the study: to assess the impact of the 5 years ZEST intervention on the genetic structure of S. haematobium was probably not achievable, in the sense that schistosomes in Pemba were exposed to a very intense drug administration for several years and therefore expecting a measurable genetic change (in very similar arms) due to an intervention that lasted 5 years i was a little bit optimistic. However the study provide important information on other aspects of schistosomiasis control and therefore merits publications.

I have only minor suggestions:

1- in my opinion in material and methods section the authors should report with a little more details the previous story of treatment with praziquantel in Pemba (that this is probably over 20 years longer). This is important because will put into prospective the activities conducted during ZEST. 

2. line 306 intensity of infection I think the author in addition to the main intensity of infection should also provide the intensity of infection by class (in order to give an idea of the changes occurred in the prevalence on these classes during ZEST).

3-Figure 2 the unit of measure (eggs / 10 ml urine) of the mean intensity of infection should be indicated in the Y axis

Discussion

4- I do not agree with the conclusion of the authors on line 438-39: “ difference in parasite fecundity also appeared to explain some of the hotspot observed” in my opinion from Figure 6ab shows that all range of parasite fecundity at baseline or follow can be present in hot spot (both in “brown” and “black” hotspot)

Conclusion

5- line 510-511 I is not clear to me on which basis the authors mention “with the possible exception of an enhanced impact of molluscicide combined with MDA” the authors suggest that molluscicide combined with MDA has impact on genetic structure? It does not seems the case from this study.. can the authors better explain?

Reviewer #2: Penance and colleagues describe a large-scale population genetic analysis of Schistosoma haematobium from two islands off the coast of Zanzibar. The rationale for the study was to examine the genetic consequence of different control strategies tested under a cluster-randomised trail, comparing ~1500 miracidia collected before and another ~1500 miracidia collected 5 years after intervention. The results presented suggests little genetic differentiation overall, but some variation in fecundity between and within islands, and some hints of genetic structure potentially indicative of sub-species variation. 

The manuscript is well written and the methodology appropriate. I think there could be some greater emphasis on showing results and data for some of the key conclusions which we lacking in places. I have made some suggestions below that are worth considering and may help the reader. Overall, this study has implications for the way in which schistosomiasis is controlled and emphasises the utility of genetics as part of the decision-making process, and will be a good fit for PLoS NTD.

I look forward to seeing a revised version. 

Kind regards, 

Stephen Doyle

Wellcome Sanger Institute

General comment

- A key result is stated in the abstract, results and discussion is that there is a trend toward reduced genetic diversity over time. However, apart from the statistical test reported in the result, there are no data that shows this either in the results or supplementary data. Given this is a population genetic analysis, at least a table with allelic richness and inbreeding coefficients could/should be shown to support these conclusions.

Specific comments

Key words

- Haematobium is spelled incorrectly.

- Could be consistent with capitalistion.

Line 180: multiplexed microsatellite PCR panels (Panel 1 and Panel 2) as described in [22] [23, 25, 26].

- Personally, I find it frustrating when reactions are not described. Could some details on the reaction setup and cycling conditions be included, even if in the supplement?

- The information in the supplement, ie some primer sequences, looks incomplete as it is. Could this be checked?

Line 196: “Actual numbers….”

- Could those numbers be described?

Line 201: “Data analyses”

- Could the code used throughout be deposited in a stable repository? 

Lines 183-186: “…cherry picked….”

- Seems like there might be some redundant information here (cherry-picked is described twice for what I think is the same thing). Worth another check.

Line 344: “sibship reconstruction”

- I was excited to see this used, but felt like there was a missed opportunity to explore / describe the actual sibships, and whether there was any difference in the degree of sibship relationships between treatment arms or over time. Could this at least be commented on?

- I appreciate the analysis does go on to estimate fecundity, but this is an average. Your genetic data (I presume) gives you precise(?) measures.

Line 438: “Differences in parasite fecundity also appeared to explain some of the hotspots observed across islands and arms (Figure 6a,b).”

- This seems like a bit of a weak conclusion from the Figure. Could this be more explicitly tested?

Line 445: “Further evidence of potential strain variations amongst parasites by both island, and potentially hotspot within islands, was revealed by PCA analyses“

- There could be a bit more discussion around what is driving this difference within islands, esp given the distribution of hotspots, ie what factors lead to a hotspot, and what might lead to strain differences. The discussion is predominantly focused on between islands, not within islands. 

- Is it not a bit curious that the variation explained by between islands in the PCA is basically the same was within island hotspot vs non-hotspot? I would have expected a stronger signal between islands than subsets of populations within islands.

Line 483: “hotspots were found to be clustered with other hotspots“

- Really? Is there a way to objectively define this?

- Cross checking the cluster of highest mean fecundity populations in Fig 6a vs Fig 7a shows they are spread all over Pemba. 

Line 485: “corroborating a recent study of S. haematobium in Zanzibar“

- Reference?

Line 514: ‘Whole Genome Sequencing“

- Doesn’t need capitalisation

Figure 1 

- It is very difficult to take anything away from this figure. It is much more like a graphical abstract rather than provide anything informative. 

- I’d suggest either removing it, or adding information, sample sizes, a decision tree to describe how these different aspects of the study relate to each other. 

- Some data from Supplementary Tables 1 and 2 might be useful here. 

Figure 4.

- Is this figure needed? Figures 5 and 6 seem to cover this pretty well. 

Figure 6

- Should there be some form of error bars on these mean estimates?

- Alternative to error bars – could it be worth combining the two plots (a+b), and use a different shape for the year? Could place a line connecting the two datapoints per site. That way, the reduction in fecundity between time points would be obvious. 

- The variance looks much lower, especially in the 5 y group, than the confidence intervals show in Figure 5. Error bars in Fig 6 might be helpful showing this. 

- Kinyasini_(Pemba) – should this be labelled in this way? Also, I couldn’t find this population on the maps in Fig 7. 

Figure 7

- It is not quite clear to me what is the rationale for using a different colour scheme between a+b and c+d. Clearly, there are big differences between yr 1 and yr 5, however, different colour schemes prevent them from being compared?

- Is the spatial distribution within a time point more important than between time points?

- Could scale bars be placed on the maps?

Figure 8

- It would be good to show the axis labels with % variance on them, rather than in the figure legend

- “d=0.5” is shown in all of the plots, but it not described anywhere. Perhaps remove from the plots, add it to the legend (if important to do so), and explain what it is.

- Please describe what the ellipse represents in the legend.

Data availability: “No - some restrictions will apply”

- Could this be explained in more detail?

PLOS authors have the option to publish the peer review history of their article (what does this mean?). If published, this will include your full peer review and any attached files.

Reviewer #1: No

Reviewer #2: Yes: Stephen R. Doyle

Figure Files:

Data Requirements:

Reproducibility:

References

---

## [Decision Letter · Decision Letter 1]

13 Sep 2022

Dear Prof. WEBSTER,

We are pleased to inform you that your manuscript 'Potential drivers for schistosomiasis persistence: population genetic analyses from a cluster-randomized urogenital schistosomiasis elimination trial across the Zanzibar islands' has been provisionally accepted for publication in PLOS Neglected Tropical Diseases.

Best regards,

jong-Yil Chai

Academic Editor

Maria Elena Bottazzi

Section Editor

The revised manuscript has been reviewed by previous reviewers. They are satisfied either with the original version or with the revised version. I agree with them.

Reviewer's Responses to Questions

**Key Review Criteria Required for Acceptance?**

**Methods**

-Are the objectives of the study clearly articulated with a clear testable hypothesis stated?

-Is the study design appropriate to address the stated objectives?

-Is the population clearly described and appropriate for the hypothesis being tested?

-Is the sample size sufficient to ensure adequate power to address the hypothesis being tested?

-Were correct statistical analysis used to support conclusions?

-Are there concerns about ethical or regulatory requirements being met?

Reviewer #2: (No Response)

**Results**

-Does the analysis presented match the analysis plan?

-Are the results clearly and completely presented?

-Are the figures (Tables, Images) of sufficient quality for clarity?

Reviewer #2: (No Response)

**Conclusions**

-Are the conclusions supported by the data presented?

-Are the limitations of analysis clearly described?

-Do the authors discuss how these data can be helpful to advance our understanding of the topic under study?

-Is public health relevance addressed?

Reviewer #2: (No Response)

**Editorial and Data Presentation Modifications?**

Reviewer #2: (No Response)

**Summary and General Comments**

Reviewer #2: Thank you to the authors for providing a thorough revision. All my comments were adequately addressed or reasonably rebutted, so I am very happy to support publication in PloS NTD. Congratulations on a great piece of work, and I am looking forward to seeing it in print.

Kind regards,

Stephen Doyle

PLOS authors have the option to publish the peer review history of their article (what does this mean?). If published, this will include your full peer review and any attached files.

Reviewer #2: **Yes: **Stephen Doyle

---

## [Editor Report · Acceptance letter]

6 Oct 2022

Dear Prof. WEBSTER,

We are delighted to inform you that your manuscript, "Potential drivers for schistosomiasis persistence: population genetic analyses from a cluster-randomized urogenital schistosomiasis elimination trial across the Zanzibar islands," has been formally accepted for publication in PLOS Neglected Tropical Diseases.

Best regards,

Shaden Kamhawi

co-Editor-in-Chief

Paul Brindley

co-Editor-in-Chief
